# A Preliminary Assessment of Horticultural Postharvest Market Loss in the Solomon Islands

**Steven Jon Rees Underhill [1,2,\*], Leeroy Joshua [2] and Yuchan Zhou [1]**

[1]    School of Science and Engineering ML41, University of the Sunshine Coast, Locked bag 4,
       Maroochydore DC, Queensland 4558, Australia; yzhou1@usc.edu.au

[2]    School of Natural Resources and Applied Sciences, Solomon Islands National University,
       PO BOX R113 Honiara, Solomon Islands; leeroy.joshua@sinu.edu.sb

[\*]    Correspondence: Sunderhi@usc.edu.au; Tel.: +61-754-565-142

**Abstract:** Honiara's fresh horticultural markets are a critical component of the food distribution system in Guadalcanal, Solomon Islands. Most of the population that reside in Honiara are now dependent on the municipal horticultural market and a network of smaller road-side markets to source their fresh fruits and vegetables. Potentially poor postharvest supply chain practice could be leading to high levels of postharvest loss in Honiara markets, undermining domestic food security. This study reports on a preliminary assessment of postharvest horticultural market loss and associated supply chain logistics at the Honiara municipal market and five road-side markets on Guadalcanal Island. Using vendor recall to quantify loss, we surveyed a total of 198 vendors between November 2017 and March 2018. We found that postharvest loss in the Honiara municipal market was 7.9 to 9.5%, and that road-side markets incurred 2.6 to 7.0% loss. Based on mean postharvest market loss and the incidence of individual vendor loss, Honiara's road-side market system appears to be more effective in managing postharvest loss, compared to the municipal market. Postharvest loss was poorly correlated to transport distance, possibly due to the inter-island and remote intra-island chains avoiding high-perishable crops. Spatial mapping of postharvest loss highlighted a cohort of villages in the western and southern parts of the main horticultural production region (i.e., eastern Guadalcanal) with atypically high levels of postharvest loss. The potential importance of market-operations, packaging type, and mode of transport on postharvest market loss, is further discussed.

**Keywords:** food security; postharvest; post-harvest; Pacific; food loss; municipal market; road-side market; Honiara; Guadalcanal; Malaita

## 1. Introduction

Solomon Islands is a South Pacific archipelago consisting of six major islands and a further 986 smaller islands, atolls and reefs. Around 84% of Solomon Islanders reside in rural villages and are dependent on subsistence-based agriculture and local fisheries [1,2]. In recent times, commercial food supply chains have become increasingly important in the Solomon Islands due to a combination of rural to urban population drift [3,4], population growth [5,6], ongoing challenges associated with agricultural productivity [7], and the impacts of adverse weather events [1,2,8]. This trend is particularly acute in the capital Honiara, with only 32% of the urban population having access to a home garden [6]. Most of the population that resides in Honiara, are now dependent on the municipal horticultural market and a network of smaller road-side markets to source their fresh fruits and vegetables.

Honiara's horticultural markets not only provide important food security and human nutrition outcomes [9,10], but create opportunities for local economic development and demonstrate a strong gender participation bias in favor of women market vendors [9,11]. The income generated from these

markets provides essential livelihood support for local squatter settlements in the "greater Honiara" region [3] and are a primary source of income for many close proximity islands such as Savo Island [7] and possibly Florida Island. This combination of socio-economic, pro-gender engagement and food security and nutrition benefits, has led to an increased focus by donors on market-based interventions in the Solomon Islands [12].

The need to improve the operational efficiency and effectiveness of the Honiara municipal market have been widely recognized [2,11,12]. The Honiara municipal market is constrained by overcrowding, poor sanitation and concerns about vendor safety [12–14]. Most studies undertaken in support of the Honiara municipal markets have done so from a community resilience, gender and human security perspective [3,4,7,11,15,16]. Its only recently that the underlying horticultural supply chains have been examined in any detail [7,11,16], providing a wider understanding of farm demographics, transport logistics and vendor practice. What remains unclear, is how efficiently the Honiara markets and their associated supply chains operate in terms of postharvest horticultural loss. Unlike other South Pacific islands such as Fiji [17,18] and Samoa [19], there are no previous reported studies on postharvest market loss in any of the markets in the Solomon Islands. With generic poor postharvest handling, potentially high-levels of postharvest loss in Honiara markets could be undermining domestic food security.

This study reports on a preliminary assessment of postharvest horticultural market loss and associated supply chain logistics at the Honiara municipal market and five road-side markets on Guadalcanal Island. The inclusion of Honiara road-side markets in this study reflects an increasing recognition of their importance in the overall food distribution system in Solomon Islands [15]. This study is part of an ongoing longitudinal assessment of postharvest horticultural loss in Honiara municipal market and road-side markets (Guadalcanal Island), Auki municipal market (Malaita Island) and the Gizo municipal market (Ghizo Island).

## 2. Materials and Methods

### 2.1. Location

This study was undertaken at the Honiara municipal market and five road-side markets in the Honiara district, Guadalcanal Island and Solomon Islands (Figure 1A,B). The location of the road-side markets assessed: Henderson, Fishing village, Lungga, King George VI and the White river, is shown in Figure 1B,C.

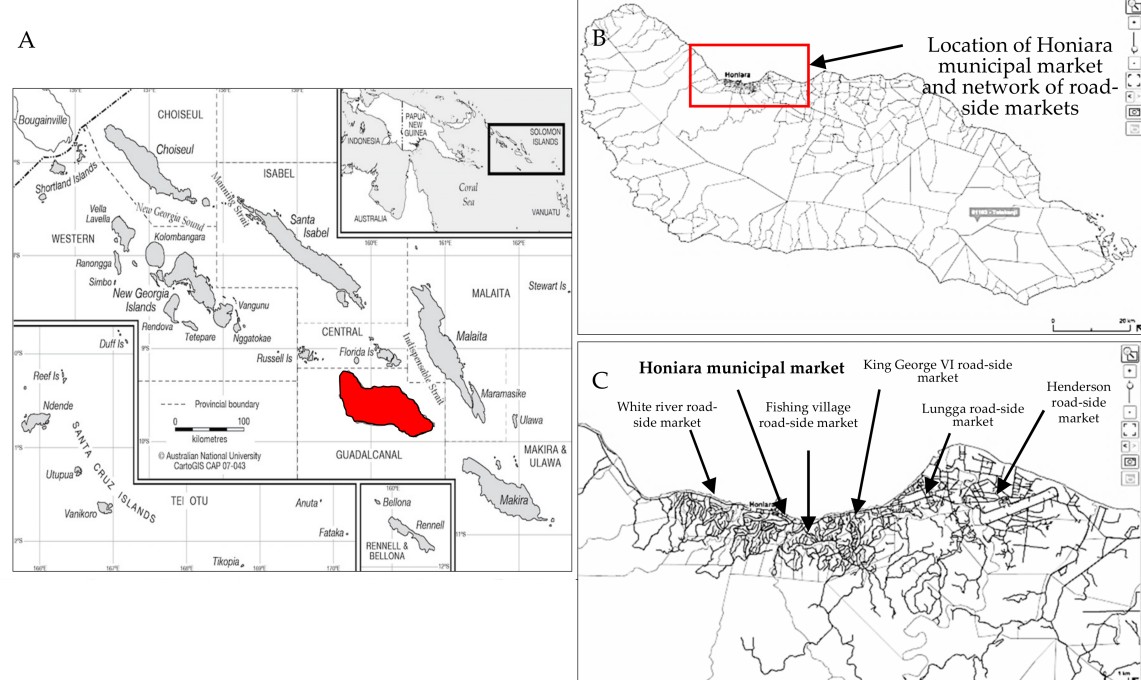

**Figure 1.** Map of the Solomon Islands. (**A**) Location of Guadalcanal Island (indicated in red) within the Solomon Islands archipelago (Map source: CartoGIS Services, College of Asia and the Pacific, The Australian National University, Australia, 2018); (**B**) map of Guadalcanal Island (red square indicates the study site); (**C**) location of the Honiara municipal market and the five road-side markets, Guadalcanal Island (Map source: Popgis@spc.int Solomon Islands National Statistics Office, Solomon Islands, 2018).

*2.2. Survey Design and Ethics Approval*

Vendor surveys were undertaken in November 2017 and March 2018. Markets were concurrently surveyed, and involved a series of enumerators from the Solomon Islands National University (SINU) to support this study. The selection of vendors to be surveyed was randomised, but excluded those vendors unable to identify where fruits and vegetables were grown (i.e., farm location) and therefore likely to be involved in inter-market trade, those vendors selling value-added or non-perishable products, and those vendors unwilling to participate in the survey. The survey design was based on semi-structured interview questions on harvesting and packaging practice, transport, market vendor practice, and postharvest loss. Enumerators received prior training in the survey methodology and ethics compliance.

A total of 198 vendors were assessed across all of the key Guadalcanal fruit and vegetable markets. This included 104 professional market vendors at the Honiara municipal market (42 vendors surveyed in November 2017 and an additional 62 vendors surveyed in March 2018). A further 94 road-side market vendors (occasional traders) were also surveyed (42 road-side vendors surveyed in November 2017 and 52 road-side vendors surveyed in March 2018). The survey was replicated across two sampling dates to partially account for potential differences in supply chain demographics and postharvest handling practice due to crop seasonality.

Surveys involved a short semi-structured interview lasting 5–10 min, commonly undertaken in the local language. All interviews were completed in compliance with the University of the Sunshine Coast Human Research Ethics Approval (A16814).

### 2.3. Data Collected

Postharvest market loss was determined using vendor recall, consistent with other recent Pacific market loss studies [18,19]. This method excludes on-farm loss, does not include consumer waste nor does it account for potential re-use of market loss for non-human consumption (i.e., product used for animal feed). For the purposes of this study, postharvest loss is defined as a fresh horticultural product that was permanently removed from the chain due to being of an unsaleable quality and not provided to others with the intent of human consumption [20]. Vendors were asked to quantify the level of postharvest loss of the main horticultural products on-display at their individual vendor stalls. This allowed for postharvest loss and handling practice to be further segregated and analysed according to crop type.

Transport distance from the farm (village) to the market was determined using Google Earth Pro™ Distance Calculator based on the most probable road transport route. Where the location of the village could not be directly identified, transport distance was calculated by cross referencing the map location given by the vendor with the nearest village. Village locations were further validated in discussions with the enumerators. For inter-island supply chains, transport distance was based on the most likely direct ferry route. For the intra-island transport supply chains that involved a combination of boat and road transport, such as those from southern Guadalcanal, transport distance was calculated based on a boat transport route from the farm to the nearest village with continuous road access to Honiara, and the most probable road transport route thereafter.

Product was identified as either fruits, vegetables, or fruits and vegetables, based on generic (non-botanical) crop classification (i.e., tomato and similar crops were classified as vegetables). Semi-processed, processed and non-horticultural commodities were excluded from this study.

### 2.4. Statistical Analysis

Data analysis was undertaken using one-way analysis of variance (ANOVA). Analysis of market vendor survey loss was undertaken using ANOVA followed by the Tukey–Kramer multiple comparison test (with consideration for uneven vendor numbers between markets). The relationship between market loss and transport distance was determined using a linear regression analysis.

## 3. Results

### 3.1. Postharvest Loss

Mean percent postharvest market loss at the Honiara municipal market was 9.5% in November 2017 and 7.0% in March 2018 (Table 1). Mean percent postharvest loss for the road-side markets in Guadalcanal was 7.9% in November 2017 and 2.6% in March 2018. The level of postharvest loss was significantly higher in the Honiara municipal markets compared to the Honiara road-side market in the March 2018 survey.

**Table 1.** Mean percent postharvest market loss for fresh fruits and vegetables sold in the Honiara municipal and road-side markets.

| Market Type and Location | Mean Percent Postharvest Loss (November 2017) | Mean Percent Postharvest Loss (March 2018) | Vendor with No Postharvest Loss (%) |
|---|---|---|---|
| Honiara municipal market | 9.5 [z] a [w] | 7.0 [x] a | 19.2 |
| Honiara road-side markets | 7.9 [z] a | 2.6 [y] b | 44.7 |

Data relates to all fruits and vegetables combined. [z] n = 42. [x] n = 62. [y] n = 52. [w] Values followed by the same letter are not statistically different at $p < 0.05$ based on Tukey-Kramer test.

The frequency of postharvest loss differed between the municipal and road-side markets (Table 1). In the municipal market, most vendors experienced some level of postharvest loss, with only 19.2% of

vendor surveyed indicating no loss (Table 1). In contrast, nearly half of the road-side vendors (44.7%) reported no postharvest loss. When road-side market vendors incurred postharvest loss, the amount of loss tended be high (often 20 to 25% loss—data not shown).

Postharvest loss for fruits was 7% to 7.6% in the municipal market and 3% to 5.2% in road-side markets (Table 2). In comparison, postharvest loss for vegetables tended to be more variable, 1.8 to 12.7%, with significantly higher postharvest loss in municipal market in the November survey (Table 2). Low, but not significant, vegetable postharvest loss observed in road-side markets in the March survey was due to fewer vendors reporting atypically high postharvest loss (data not shown).

**Table 2.** Mean percent postharvest market loss for fresh fruits and vegetables [z] sold in the Honiara municipal and road-side markets.

| Market Type and Location | Mean Percent Postharvest Loss (November 2017) | | Mean Percent Postharvest Loss (March 2018) | |
|---|---|---|---|---|
| | Fruit | Vegetable [y] | Fruit | Vegetable [y] |
| Honiara municipal market | 7.0 efgh [x] | 12.7 abcde | 7.6 defgh | 8.1 cdefgh |
| Honiara road-side markets | 5.2 fgh | 11.6 bcdef | 3.0 gh | 1.8 h |

[z] Postharvest loss data relates to fresh fruits and vegetables but excludes all other food categories including semi-processed and cooked product. [y] Crops were defined as vegetables based on a commercial rather than botanical classification (i.e., tomato identified as a vegetable crop). [x] Values followed by the same letter within columns and rows for individual market survey dates are not statistically different at $p < 0.05$ based on Tukey–Kramer test.

The portion of fruits to vegetables being sold differed during the two survey dates, possibly reflecting seasonal supply. In November, 44% of vendors were selling fruits and 56% selling vegetables, whereas in the March survey 30% of vendors were selling fruits and 70% vegetables (data not shown).

Mean postharvest loss for inter-island and intra-island supply chains supplying the Honiara municipal market (November 2017 and March 2018 combined results) is shown in Table 3. While inter-island chains appear to have slightly higher loss, this trend could not be statistically assessed due to the limited number of inter-chains included in the survey.

**Table 3.** Mean percent postharvest market loss for intra-island and inter-island located farms supplying the Honiara municipal market.

| Supply Chains | Mean Percent Postharvest |
|---|---|
| Guadalcanal Island to Honiara (intra-Island) | 8.1 [z] |
| Malaita Island to Honiara | 16.7 [y] |
| Savo Island to Honiara | 11.2 [x] |
| Nggela Island to Honiara | 16.3 [w] |

[z] n = 90 [y] n = 3; [x] n = 4; [w] n = 2.

*3.2. Supply Chain Logistics*

Fresh fruits and vegetables sold in the Honiara municipal market were primarily sourced from farms located to the east of Honiara, and to a lesser extent, villages on the north–west of Guadalcanal Island (Figure 2). Products sourced from farms located to the west of Honiara were more common during the November sampling period. Few farms located in the southern parts of Guadalcanal supply the Honiara municipal market. A small percentage of Honiara municipal market vendors (8.7%) were sourcing produce from Malaita, Gizo and Savo Islands (Figure 2). Inter-island sourced products were only observed in the Honiara municipal market, with the road-side markets tending to source locally-grown products.

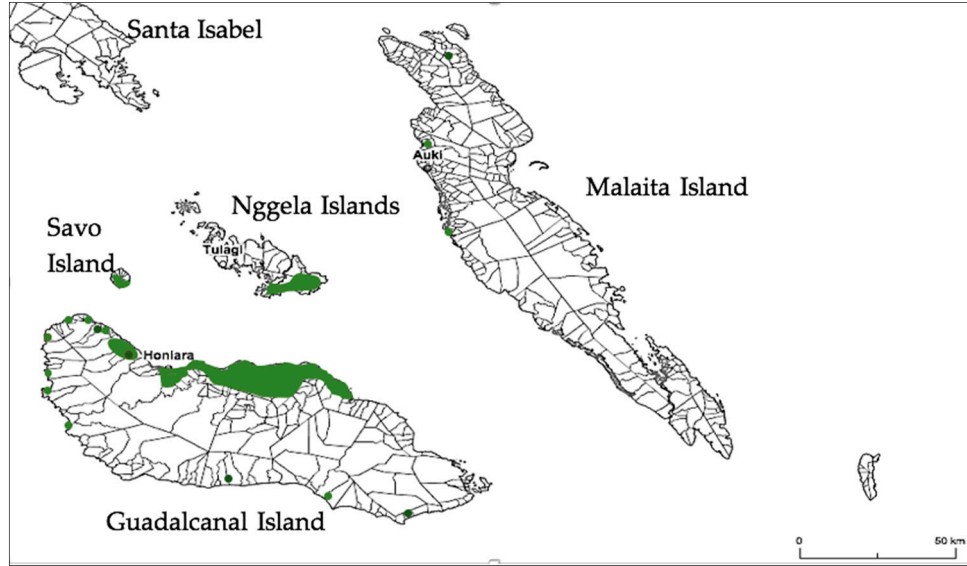

**Figure 2.** The locations (green marked areas) of farms supplying the Honiara municipal and road-side markets during the survey period (November 2017 and March 2018 data combined). (Source: Base map: Popgis@spc.int Solomon Islands National Statistics Office, Solomon Islands, 2018). Note location of farms are not GIS positioned.

Horticulture transport logistics into the Honiara municipal market were relatively short, with products travelling 40 to 47 km (Table 4). In comparison, products supplying the road-side markets travelled 19 to 27 km, almost half the distance. Some of this disparity can be attributed to the inclusion of inter-island supply chains into the Honiara municipal market. When the median transport distances are considered, the transport distance between farms and municipal markets or road-side markets were relatively similar in the November 2017 survey. In the March 2018 survey, mean transport supply distance for road-side markets was 17.1 km (Table 4). This reduction in transport distance implies vendors are able to source more products locally, and may explain the lower incidence of postharvest loss observed in road-side markets during this time (Table 2).

**Table 4.** Transport distance from the farm to the municipal or road-side markets.

| Market Type and Location | Mean Transport Distance (km) | Median Transport Distance (km) |
|---|---|---|
| Honiara municipal market (November 2017) | 40.0 | 32.9 |
| Honiara road-side markets (November 2017) | 26.9 | 28.0 [z] |
| Honiara municipal market (March 2018) | 46.6 | 38.1 |
| Honiara road-side markets (March 2018) | 18.6 | 17.1 [z] |

[z] Road-side market data represents data sourced from the Henderson, Fishing Village, Lungga, King George VI and White river road-side markets.

The mean transport distance for the individual road-side market network varied depending on the market location and the survey date (Table 5). Products sold at the Lungga and King George VI markets tended to be sourced from smallholder farmers located in close proximity to these markets (1 to 2 km away). Whereas products supplying the larger White river and Fishing village markets travelled 24 to 37 km. The comparatively shorter transport distances for the White river noted in the November survey and for the Henderson and Fishing village markets in the March survey are thought to reflect possible crop seasonal variability in the supply chains.

**Table 5.** Mean transport distance from the farm to the individual road-side markets.

| Market Type and Location | Mean Transport Distance (km) (November 2017) | Mean Transport Distance (km) (March 2018) |
|---|---|---|
| Henderson | 40.7 | 14.3 |
| Lungga and King George VI (combined) | 2.16 | 1.35 |
| Fishing Village | 37.7 | 26.8 |
| White river | 24.3 | 30.9 |

The most common mode of transport used by farmer/vendors to transport product to the Honiara markets (municipal and road-side) was by truck (Table 6). Truck-based transport systems were associated with farms located in more remote intra-island locations, with a mean travel distance of 37 km. However, there was considerable variability in transport distances involving trucks, with the shortest recorded transport distance being 6.2 km and the furthest being 64.8 km.

**Table 6.** Mode of transport used and mean transport distance for all markets and all survey dates.

| Mode of Transport | Mean Transport Distance (km) | Percent of Farmers/Vendors Using Specific Mode of Transport (%) |
|---|---|---|
| Ferry/boat | 88.9 a [z] | 6.7 |
| Truck | 37.0 bcde | 54.2 |
| Car | 25.2 cde | 4.5 |
| Minivan/public bus | 20.5 de | 14.5 |
| Taxi | 8.5 e | 13.4 |
| Walk | 1.3 f | 6.7 |

[z] Values followed by the same letter are not statistically different at $p < 0.05$ based on Tukey–Kramer test.

Mean transport distance involving cars or minivans/public buses was 20 to 25 km (Table 6). There was also considerable variability in the transport distance by car—ranging from 3.7 to 44.7 km, and transport distance by minivan/public bus—ranging from 0.5 to 41.2 km.

Transport by taxi was limited to farmers located relatively close to the market, with a mean transport distance of 8.5 km (Table 6).

*3.3. Potential Contributions to Postharvest Loss*

There was a weak correlation between transport distance and postharvest loss (Figure 3). Farms with very high levels of postharvest loss (>30% loss) were primarily located within 50 km of the markets. Conversely, most supply chains with a transport distance of 100 to 200 km had less than 10% loss.

The location of farms with moderate (10 to 19%) to very high levels (>30%) of postharvest horticultural loss are shown in Figure 4. Elevated postharvest loss was more prevalent in supply chains sourcing products from the far eastern part of the main production center (see Figures 2 and 4). There were multiple supply chains sourcing products from Tutumu, Tenaru, Vatukukau, Ruavatu, Siara, Binu, Aola, Tasimboko, Dadai villages on Guadalcanal Island, and Matakwara and Buma villages on Malaita Island with moderate to very high levels of postharvest loss. While there are relatively few farms located on the southern and far western parts of Guadalcanal supplying the Honiara markets (Figure 2), none of these had elevated postharvest loss (Figure 4).

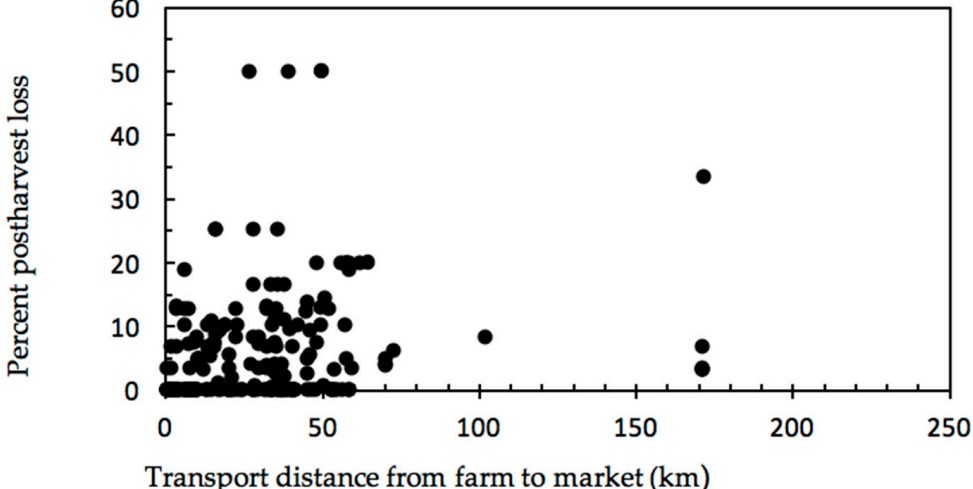

**Figure 3.** A linear regression analysis of percent postharvest loss verses transport distance for all vendors, markets and survey dates (n = 346). $R^2$ = 0.2503.

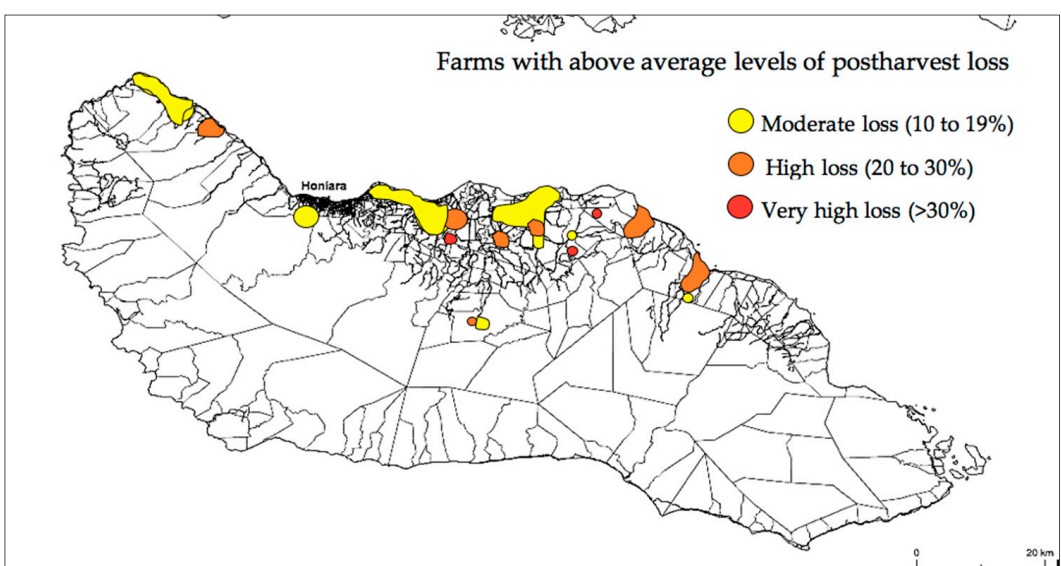

**Figure 4.** The locations of farms supplying the Honiara municipal or road-side markets with elevated levels of postharvest loss. (Source: Base map: Popgis@spc.int Solomon Islands National Statistics Office, Solomon Islands, 2018). Note farm locations are not GIS positioned.

The type of products being sourced by market vendors differed depending on farm location (Figure 5). Inter-island supply chains and those chains sourcing from the remote farms on Guadalcanal Island were less likely to include vegetables. Vendors instead tended to source vegetables from closer proximity intra-island located farms, especially those in the "greater Honiara" region and north-eastern Guadalcanal.

The most commonly sourced product from remote farms (>50 km) was watermelon, green banana and English cabbage (Table 7). Highly-perishable crops sourced from remote farms on Guadalcanal tended to be higher-value Asian leafy vegetables such as Pak choi and Choy sum (Table 7). Mean postharvest loss for these chains was 13.2% with half the consignments incurring ≥20% loss (data not shown).

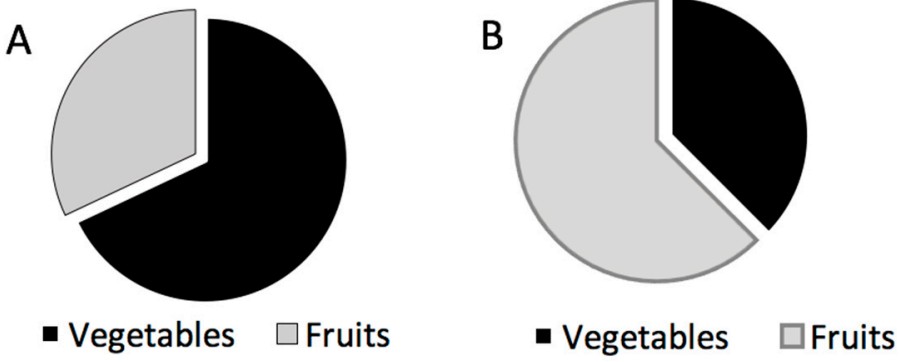

**Figure 5.** The commodity composition (vegetables to fruits ratio) of consignments sourced from intra verses inter-island located farms. (**A**) Intra-island supply chains (Guadalcanal) into the Honiara market; (**B**) Inter-island supply chains into the Honiara market. Data is based on number of consignments, rather than consignment volume or weight.

**Table 7.** The most common commodities being sourced by vendors at the Honiara municipal market from remote located farms (>50 km from farm to market).

| Commodity | Rank Order |
|---|---|
| Watermelon | 13.3% |
| Green banana, English cabbage | 11% |
| Pak choi, pineapple | 8.9% |
| Cucumber, shallots | 6.7% |
| Choy sum, citrus | 4.4% |

A wide range of different packaging types were observed in the markets (Table 8). Large sacks (≥40 kg) were the most common type of packaging, especially for leafy indigenous vegetables. Higher value crops such as tomato and Asian vegetables tended to be limited to smaller (<20 kg) packing units. Postharvest loss was highest in very large packing units (Table 8).

**Table 8.** Mean postharvest loss based on packaging type.

| Package Type | Mean Postharvest Loss (%) | Percentage of Supply Chains Using Packaging Type [y] |
|---|---|---|
| Very large sacks (>100 kg net weight) | 10.9 a [z] | 8.4% |
| Large sacks (approx. 40 kg) | 4.9 b | 34.0% |
| Medium sacks (20 kg) | 5.5 ab | 17.9% |

[z] Values followed by the same letter are not statistically different at $p < 0.05$ based on Tukey–Kramer test. [y] Vendors also used a range of other packaging options: plastic trays (14.2% of vendors), small plastic bags (5–10 kg) (10.5%), plastic crates (1.5%), plastic buckets (3%), steel basins (8.4%), locally woven baskets (1.5%) and nil packaging (1.5%).

## 4. Discussion

Horticultural postharvest loss in the Honiara municipal market was 7.9 to 9.5%. In comparison, postharvest loss in the Honiara road-side markets tended to be lower (2.6 to 7.0%) but more variable. This level of loss was consistent with other South Pacific municipal markets, with Reference [19] reporting a 6.2% loss in the central municipal market in Samoa. Most municipal market vendors in Honiara experienced some level of postharvest loss, whereas road-side market vendor loss tended to be less common. Based on mean postharvest market loss and the incidence of individual vendor loss, Honiara's road-side market system appears to be more effective in minimising postharvest loss, compared to the municipal market.

The potential contributors to postharvest market loss in Guadalcanal markets and reasons for reduced loss in the road-side markets are likely to be multifaceted. Diverse market participation (commercial-scale farmers through to semi-subsistence farm surplus), poor road infrastructure, the lack of a cool chain, limited or poor packaging, and inadequate market storage facilities needs to balance against potential supply chain practices that seek to mitigate or lessen potentially elevated postharvest loss. While the contributors to generic postharvest loss in horticultural markets have been widely reported [19,21–24], the inclusion of possible vendor or farmer strategies to reduce this loss are often overlooked.

Intuitively, it would be logical to assume that transport distance would have a significant effect on the level of postharvest loss seen in the market, consistent with the findings in other postharvest supply chain studies [24,25]. While inter-island supply chains appear to have higher levels of postharvest loss compared to intra-island chains, we found that postharvest loss was poorly correlated to transport distance. Farms with very high levels of postharvest loss (>30% loss) tend to be located within 50 km of the markets, most supply chains with a transport distance of greater than 100 km have less than 10% loss, and loss associated with very remote intra-island supply routes was similarly less than 10%. These observations would suggest that the distance horticultural produce needs to travel from the farm to market is not a good indicator of potential market postharvest loss in Guadalcanal.

The type of crops sourced from inter-island and remote intra-island farms and their associated supply chain practice may provide some insight into the disconnect between transport distance and postharvest loss. Most inter-island supply chains included in this study were dominated by semi-perishable crops such as watermelon, pineapple and citrus. Such crops are often considered to be more tolerant of challenging transport logistics and potentially prolonged market storage. In the more remote Malaita to Guadalcanal inter-island supply chains, the product was sourced from two fruit production centers, watermelons from Buma and pineapples from Bina. These chains involved commercial-scale farms with relatively predictable transport logistics, with resultant postharvest loss being relatively low (<5%). Georgeou et al. [11] reported that the most commonly traded crops from Savo and Nggela Islands into the Honiara markets were fruits, nuts and root crops. In remote intra-island chains, such as products sourced from Mbalo on the far south-eastern part of Guadalcanal and Tangarare on the far south eastern part of Guadalcanal, there was a similar dominance of semi-perishable crops such as watermelon and citrus. While this might simply reflect local agronomic production conditions favouring certain crops, it is also possible that there is deliberate strategy by farmers supplying the Honiara market to avoid highly perishable cash crops if the associated transport logistic is likely to incur high-levels of postharvest loss.

Vegetable supply chains still represented a significant portion of the overall inter-island trade into Honiara. A recent study of the Savo to Honiara market supply chains [16] reported not only semi-perishable crops but also highly perishable leafy vegetables being traded. Savo farmers indicated high levels of postharvest loss due to in-transit damage and delays in accessing transport [16], even though Savo Island is only about 35 km from Honiara. The presence of inter-island trade of perishable vegetable crops in spite of high-levels of postharvest loss is interesting. Georgeou et al. [16] reported that much of the trade from Savo Island into the Honiara municipal market was due to opportunistic market participation due to surplus local production [16]. Faced with possibly few alternative local market opportunities on Savo Island, potentially high postharvest loss does not appear to disincentivise market participation.

When intra-island vendor loss was analysed in terms of where produce was grown, we found that there was a cohort of villages in the western and southern parts of the main horticultural production (which is located in eastern Guadalcanal) which were consistently associated with atypically high levels of postharvest loss. This result might reflect the type of crops grown in these locations, with Reference [11] reporting that most of the perishable leafy vegetables sold in the Honiara municipal market were sourced from farms located in north-eastern Guadalcanal. An alternative or additional possibility is a lack of reliable commercial transport options in these villages, or generic poor harvesting

and handling practice. Further studies are required to better understand on-farm postharvest practice and supply chain logistics within these villages. Spatial mapping of high-loss postharvest chains has not been previously reported in the South Pacific, and provides useful information in terms of helping to better target possible future technical farmer assistance and supply chain remediation.

Supply chain modes of transport associated with Honiara's markets reflect the diversity of agronomic production systems, from the commercial-scale through to semi-subsistent trade farm surplus. The most common form of transport was open trucks, consistent with the findings reported by Reference [11]. Nearly all of the supply chains sourcing products from eastern Guadalcanal were dependant on trucks, possibly reflecting the volume of trade, poor road conditions and some level of local transport coordination. In Samoa and Vanuatu, where there is a relatively well maintained road-network and small production volumes, public buses, minivans and private vehicles are more commonly used [19]. While the mode of transport is interesting, the specific postharvest transport conditions need to be better understood. How crops are loaded and the load configuration within the truck, the volume being transported, other possible items being co-transported can also have a significant influence on postharvest loss. More work is required to better understand transport logistics especially between eastern Guadalcanal and the Honiara markets as a possible contributor to postharvest loss.

A range of packing types were used by farmers, the most common of which was 40 kg of woven sacks. Given the large diversity of crops and packaging options, only a superficial assessment of the implication of packaging type on loss could be undertaken. As anticipated, very large agricultural sacks (>100 kg) used transport traditional leafy vegetables incurred significantly high levels of postharvest loss compared to smaller sizes of the same packaging type. Most heavy produce (such as pineapples, watermelon) were transported loose (no packaging). In the case of pineapples, the product was often tied into bundles of up to 40 fruit and carried using wooden poles. Plastic crates were rarely observed. Plastic buckets and steel trays were used for crops prone to damage during transport (such as tomato and papaya). The packing options used by farmers and vendors is thought to simply reflect the type of packaging readily available, with Reference [23] noting that vendors in Malaita Island were aware of the adverse implication of poor packaging.

Comparatively low postharvest loss (4 to 5%) associated with a commonly used form of packaging (i.e., woven sacks ≤40 kg) would suggest that while packing is far from ideal, for most farmers packing had little effect on resultant postharvest loss. However, damage associated with poor packaging can be latent and, therefore, not immediately evident when product arrivals at the market. Georgeou et al. [11] reported that product in the Honiara municipal market is commonly sold with 1/2 to 1 day or arriving at the market. It is possible that the potential full implications of poor packaging may be somewhat negated due to rapid market-throughput.

How efficiently the market-to-consumer food system operates directly influences postharvest supply chain loss. Noting high tropical ambient conditions, prolonged market storage has been reported to significantly elevate postharvest loss in other Pacific horticultural markets [19]. The observation by Reference [11], that most vendors in the Honiara municipal market sell their produce within 1/2 to 1 day is therefore significant. Honiara's road-side markets are likely to experience even more rapid product throughput due to fewer vendors and smaller volumes of product being sold, reducing vendor competition, and road-side markets located close to the resident's areas increasing potential consumer accessibility. In comparison, a product traded through the municipal market in Samoa is often stored for 2 to 3 days before it can be sold [19]. In Samoa, the benefits of comparatively good on-farm postharvest handling practice and shorter transport distances are being undermined by prolonged market storage [19]. In the Honiara markets, rapid market throughput of a perishable product is thought to be an important factor in avoiding potentially higher-levels of postharvest loss due to poor on-farm and transport practice. Fast on-selling by vendors in the Honiara municipal market is not the result of a better designed market infrastructure. Instead, high market vendor fees, over-crowding, poor market storage conditions, and significant concerns over vendor safety and

hygiene create tangible incentives for Honiara vendors to sell their produce as quickly as possible. Further studies are required to better understand road-side market trading practices and whether this further contributes to slightly lower postharvest loss in these markets. The implications of current vendor practice on postharvest loss at the consumer-end of the value chain also warrants further investigation.

One variable that needs to be considered when interpreting market survey data in this study is the potential for inter-market trade (particularly between the Honiara municipal market and the various road-side markets). Georgeou et al. [11] reported that approximately 30% of consumers at the Honiara market were on-selling products in other markets. In this study, we sought to exclude vendors who had sourced products from other markets from the survey, however, 2.6% of market vendors surveyed were unable to identify the farm location where the product was sourced. However, given that Reference [11] further highlighted ongoing tension between farmer vendors and re-sellers, suggesting that re-sellers may not self-identify when surveyed, we cannot exclude the possibility of some level of data error based on vendors providing deliberately inaccurate survey responses.

## 5. Conclusions

Horticultural postharvest loss in the Honiara municipal market is consistent with the level of loss in the Apia municipal market, Samoa. Guadalcanal's road-side vendors appear to experience less postharvest loss than vendors in the municipal market; however, the reasons for this are still unclear. The level of loss observed in Guadalcanal's postharvest markets is thought to be due to a combination of poor packaging, the type of crops being sold and possible opportunistic market participation associated with trade farm surplus. While the types of transport used by smallholder farmers were documented, their contribution to postharvest loss is unclear. We believe that potentially higher market loss is being mitigated by market vendor practice. Rapid market throughput-associated fast on-selling of the product reduces the time a product requires to be stored in the market. Farmers with potentially challenging transport supply chain logistics, which are likely to incur high postharvest loss, appear to be avoiding highly perishable crops in favor of semi-perishable fruit and starchy root crops. The observation of a series of farms toward the western and southern margins of the main production center with atypically high levels of postharvest loss warrants further investigation. Similarly, further work is required to better understand on-farm harvest and postharvest practices and possible elevated loss at the consumer-end of the chain.

**Author Contributions:** Conceptualization, S.J.R.U. and L.J.; methodology, S.J.R.U. and L.J.; investigation, L.J.; statistical analysis, Y.Z.; writing—original draft preparation, L.J. and S.J.R.U.; writing—review and editing, S.J.R.U., L.J. and Y.Z.; supervision, S.J.R.U.

**Funding:** This research was funded by the Food and Agriculture Organisation of the United Nations (FAO) grant number LoA SAP 2017/16 "Policy measures for the reduction of food loss/waste along fruit and vegetable value chains".

**Acknowledgments:** We would like to express our sincere appreciation for the invaluable assistance and support provided by Michael Ho'ota and Selson John Ulasi (Ministry of Agriculture and Livestock, Solomon Islands), Nichol Nonga (FAO), Peter Iro (Solomon Islands National), and the Late Tim Martyn (FAO). We would also like to acknowledge students from the Solomon Islands National University who assisted in data collection and the numerous Solomon Island road-side and municipal market vendors and small-holder farmers, who provided their time and input in support of this study.

**Conflicts of Interest:** The authors declare no conflict of interest. The funders had no role in the design of the study; in the collection, analyses, or interpretation of data; in the writing of the manuscript, or in the decision to publish the results.

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
