# Peer review of "A Preliminary Assessment of Horticultural Postharvest Market Loss in the Solomon Islands"

_horticulturae, doi:10.3390/horticulturae5010005_

Round 1
Reviewer 1 Report
The authors must be complimented on a well written manuscript. Insight is provided with the detailed methodology. The discussion answered the objectives. Apart from a few errors as seen in the attached manuscript I recommend that it be allowed to be published. I have a slight reservation on the direct writing in the results. I would prefer indirect writing.

Author Response
Response to Reviewer 1
· We would like to retain the current title, “A preliminary assessment ….”, because this is the first study of postharvest loss in Solomon Islands and whilst the data is of critical important, further work is required to validate and prioritise possible causes of observed postharvest loss.
· The reviewer is seeking an explanation as to why road-side loss is lower than municipal market loss. While we have amended the paper to highlight possible contributors, it is important to highlight that the critical finding is that a difference seems to exist (but this needs to be further verified).
· There is only one municipal market in Honiara hence our reference to municipal (not municipal markets as requested by the reviewer).
· Most authors in food loss refer to “postharvest loss” not “postharvest losses” as requested by the reviewer. We have therefore retained the former.
· All other amendments addressed as requested.

Reviewer 2 Report
Abstract and introduction does not clearly bring out the need for the study (justification of the study).
In the methodology section, there need to describe the methodology that was used to arrive at the sample size used by the authors.
Author Response
Response to Reviewer 2 (THIS REVIEWER)
· All requested amendments actioned.
· The abstract has been adjusted to better reflect the reason for the study (as requested by the reviewer)
· The methodology has been adjusted to insert new text to better articulate sampling.

Reviewer 3 Report
Postharvest losses are of major concern in horticulture supply chains. This paper gives valuable insights. However, I have some minor comments that I feel need to be addressed before the paper is accepted for publishing.
Methodology
In Section 2.2 (lines 84-98) is will be good if the authors can give a brief description of (i) how were the respondents selected and the breakdown of the vendors according to type of fruit/vegetables, (ii) how many questions and type of questions?
The information shown in the results: Table 2 and lines 144-152, is not provided in the methodology section. I suggest that the methodology be details so that it will be easy to comprehend the results.
Line 92-94: were the 198 vendors different, or some interviewed in November 2017 were also interviewed in March 2018?
Discussion
Line 285, what exactly about the road-side market could have been a contributing factor to the low postharvest losses?
Line 399: "Guadalcanal’s postharvest markets is thought to be due to a combination of poor packaging, the type of crops....". How did these characteristics of the markets compare with road-side markets since the road-side markets have lower postharvest losses?
Author Response
Response to reviewer 3 (THIS REVIEWER)
· All requested amendments actioned
· Section 2.2 has been amended to insert new text to better articulate survey design methodology.
· New text has been inserted into the methodology to better explain how crop-specific data was obtained.
· Methodology has been reworded to better explain the sampling numbers between the two survey date.
· The reasons for the differences in loss between road-side markets and the municipal market are not fully understood, moreover, further surveys are required this trend is consistent and not seasonal. We have reworded the discussion to make this clear.
